Microbial biogeography of the wombat gastrointestinal tract

Eisenhofer Raphael raph.eisenhofer@gmail.com raphael.eisenhoferphilipona@adelaide.edu.au 1 2
D’Agnese Erin 3 4
Taggart David 5 6
Carver Scott 7
Penrose Beth 3
1 School of Biological Sciences, University of Adelaide , Adelaide , South Australia , Australia
2 Australian Research Council Centre of Excellence for Australian Biodiversity and Heritage, University of Adelaide , Adelaide , South Australia , Australia
3 Tasmanian Institute of Agriculture, University of Tasmania , Hobart , Tasmania , Australia
4 School of Marine and Environmental Affairs, University of Washington , Seattle , WA , United States of America
5 School of Animal and Veterinary Sciences, University of Adelaide , Adelaide , South Australia , Australia
6 FAUNA Research Alliance, Institute for Land, Water and Society , Kahibah , New South Wales , Australia
7 Department of Biological Sciences, University of Tasmania , Hobart , Tasmania , Australia
Gillespie Joseph
Electronic publication date: 2022 Feb 23
Publication date: 2022
Volume: 10
Electronic Location ID: e12982
Received 2021 Sep 15; Accepted 2022 Feb 1
Copyright: ©2022 Eisenhofer et al.
Copyright year: 2022
Copyright holder: Eisenhofer et al.
License: This is an open access article distributed under the terms of the Creative Commons Attribution License, which permits unrestricted use, distribution, reproduction and adaptation in any medium and for any purpose provided that it is properly attributed. For attribution, the original author(s), title, publication source (PeerJ) and either DOI or URL of the article must be cited.
License URL: https://creativecommons.org/licenses/by/4.0/

Keywords: Microbial Ecology, Marsupials, Microbiome, Biogeography, Gastrointestinal, Gut, Zoology, Wombat, Australia

Funding: Australian Research Council Centre of Excellence for Australian Biodiversity and Heritage (CABAH) Australian Research Council Linkage Project LP180101251 Schultz Foundation (Raphael Eisenhofer/David Taggart) Raphael Eisenhofer was funded by the Australian Research Council Centre of Excellence for Australian Biodiversity and Heritage (CABAH). The study received funding from the Australian Research Council Linkage Project (LP180101251) (Scott Carver), and the Schultz Foundation (Raphael Eisenhofer/David Taggart). The funders had no role in study design, data collection and analysis, decision to publish, or preparation of the manuscript.

==============================
Most herbivorous mammals have symbiotic microbes living in their gastrointestinal tracts that help with harvesting energy from recalcitrant plant fibre. The bulk of research into these microorganisms has focused on samples collected from faeces, representing the distal region of the gastrointestinal (GI) tract. However, the GI tract in herbivorous mammals is typically long and complex, containing different regions with distinct physico-chemical properties that can structure resident microbial communities. Little work has been done to document GI microbial communities of herbivorous animals at these sites. In this study, we use 16S rRNA gene sequencing to characterize the microbial biogeography along the GI tract in two species of wombats. Specifically, we survey the microbes along four major gut regions (stomach, small intestine, proximal colon, distal colon) in a single bare-nosed wombat (Vombatus ursinus) and a single southern hairy-nosed wombat (Lasiorhinus latifrons). Our preliminary results show that GI microbial communities of wombats are structured by GI region. For both wombat individuals, we observed a trend of increasing microbial diversity from stomach to distal colon. The microbial composition in the first proximal colon region was more similar between wombat species than the corresponding distal colon region in the same species. We found several microbial genera that were differentially abundant between the first proximal colon (putative site for primary plant fermentation) and distal colon regions (which resemble faecal samples). Surprisingly, only 10.6% (98) and 18.8% (206) of amplicon sequence variants (ASVs) were shared between the first proximal colon region and the distal colon region for the bare-nosed and southern hairy-nosed wombat, respectively. These results suggest that microbial communities in the first proximal colon region—the putative site of primary plant fermentation in wombats—are distinct from the distal colon, and that faecal samples may have limitations in capturing the diversity of these communities. While faeces are still a valuable and effective means of characterising the distal colon microbiota, future work seeking to better understand how GI microbiota impact the energy economy of wombats (and potentially other hindgut-fermenting mammals) may need to take gut biogeography into account.

Introduction

Gut functionality and health relies on a diverse community of microbes inhabiting the entire gastrointestinal tract. These microbes are instrumental in digesting food, producing nutrients for the host, and as a potential first barrier to invading pathogens (reviewed in Rowland et al., 2018; Ubeda, Djukovic & Isaac, 2017). In herbivores specifically, gut microbes break down cellulose and other plant cellular components so they may be used as nutrients by the animal (Flint et al., 2012), and can detoxify plant defense compounds (Dearing & Kohl, 2017). Without these microbes there would be little to no nutritional value in much of the plant matter herbivorous mammals are known to consume and thrive on. In ruminants, the bulk of microbial digestion and fermentation occurs in a highly modified foregut (rumen) where nutrients are broken down by rumen microbiota and made ready for absorption through the small intestine (Immig, 1996). However, in herbivores without a rumen, the microbial breakdown of plant matter into usable nutrients and energy occurs in the latter parts of the gut—principally the cecum and colon (Stevens & Hume, 1998). Prior research has found that microbial communities vary along the mammalian GI tract (Donaldson, Lee & Mazmanian, 2015; Costa et al., 2015; Ericsson et al., 2016; Kelly et al., 2017; Crespo-Piazuelo et al., 2018; Eckburg et al., 2005; Flynn et al., 2018; Vasapolli et al., 2019). In mice, large differences were observed between cecum, distal colon, and faeces microbial communities (Gu et al., 2013; Pang et al., 2012). Marked microbial community differences between caecal, colon, and faecal samples were also identified in a study on koalas, which are in the same family as wombats (Barker et al., 2013). How representative faecal samples are to other parts of the GI tract is likely host species specific, resulting from variations in GI morphology and function. Yet our understanding of these differences outside of domesticated placental mammals remains unstudied. As a major goal of gut microbiome research is to understand how microbes influence host health and ecology, efforts should be made to better characterize the site-specific nature of microbes along the GI tract and determine how representative faecal samples are.

Wombats represent a useful herbivorous hindgut fermenting group to contribute to knowledge on microbial community differences along the GI tract. As marsupials, they are a valuable taxonomic group to add to a growing body of GI tract microbiota research. Bare-nosed (Vombatus ursinus, hereafter BNW) and Southern hairy-nosed (Lasiorhinus latifrons, hereafter SHNW) wombats represent two of the three extant wombat species and are also in the same sub-order as the koala (Diprotodontia). The BNW and SHNWs are large (20–35 kg), fossorial, metabolically depressed, nocturnal, and largely solitary marsupials found allopatrically across large areas of southeastern and southern Australia (Wells, 1989). These two wombat species also represent temperate and arid adapted grazing diets, respectively (Wells, 1989; Camp et al., 2020), and are thought to have shared a common ancestor ∼8 million years ago (Mitchell et al., 2014). The SHNW has a comparatively longer distal colon, with the BNW possessing a wider digestive tract and greater proximal colon surface area (Fig. 1) (Barboza & Hume, 1992a; Hume, 1999). Both species have caeca that are smaller and less functional than other hindgut fermenters, and the digesta passage rates are thought to be the same between wombat species (Hume, 1999). Previous research measuring short chain fatty acids indicates that most plant fermentation occurs in the proximal colon of both wombat species rather than in other regions of the GI tract (Barboza & Hume, 1992b). It has been estimated that plant fermentation accounts for >60% of a wombat’s daily energy intake (Barboza & Hume, 1992b). To date, gut microbiota research in wombats has focused on faecal samples obtained from captive (Shiffman et al., 2017) and wild wombats (Eisenhofer, Helgen & Taggart, 2021; Weiss et al., 2021), with nothing known about how microbial communities are structured throughout the GI tract, and how representative faeces are to other GI regions.

Figure 1 Illustration of wombat gut tracts and samples collected.

Drawn illustrations of Vombatus ursinus (top) and Lasiorhinus latifrons (bottom) gastrointestinal tracts based on Barboza & Hume (1992a). Stars represent locations where duplicate digesta samples were collected for the study. BNW, Bare-nosed wombat; SHNW, Southern hairy-nosed wombat.

Here, we used 16S rRNA gene sequencing to investigate the microbial community changes along the GI tract of two wombat individuals, a wild BNW and SHNW. We predicted that the proximal colon, the putative primary site of plant fermentation in wombats, would harbor a distinct microbial community to that of the rest of the gastrointestinal tract, including the distal colon in these wombat species.

Materials & Methods

Sample collection

Sample collection occurred during necropsies on two deceased wild wombats (n = 2), one BNW and one SHNW. A 17.1 kg female BNW was hit by a vehicle in the New Norfolk area, Tasmania, Australia and taken to Bonorong Wildlife Sanctuary. Injuries to this wombat were too severe for rehabilitation, and it was euthanized on 13th November 2019. Immediately following euthanasia the carcass was placed in a −20 °C freezer at the University of Tasmania, Hobart. The SHNW was captured near Swan Reach in the Murraylands of South Australia, approximately 100 km north-east of Adelaide, as part of an ecological study on seasonal reproduction and breeding, but then died later of an upper respiratory tract infection. The carcass was driven to Adelaide and put into a −20 °C freezer within 2 h of death. Given that storage of faecal samples for extended periods at room-temperature can result in microbial ‘blooming’ and distorted taxonomic compositions, the rapid freezing of the samples in this study (both within 8 h of death) should help mitigate such biases (Song et al., 2016; Lauber et al., 2010). Regardless, any biological differences between wombat species should outweigh any technical biases introduced during sample storage (Song et al., 2016; Lauber et al., 2010).

Prior to necropsy, carcasses were put into a 4 °C refrigerator and defrosted for ∼48 h. In human faecal samples, freeze-thawing was only found to influence taxon abundance after the fourth cycle of freeze-thaw (Gorzelak et al., 2015). The entire digestive systems were then removed from the abdominal cavity and the functional sections of the GI tract were labeled as per (Barboza & Hume, 1992a). For the stomach, small intestine, and proximal colon, digesta samples were collected in the middle of each functional section (Fig. 1). For distal colon samples, the BNW sample was collected halfway through the distal colon. For the SHNW four samples were collected: DC1 was collected 1 meter down from the proximal colon/distal colon boundary, DC2 was collected 2 m down, etc. Samples of the digesta were collected in duplicate in the SHNW and in triplicate in the BNW. While faecal samples could not be collected from the euthanized animals, the distal colon samples collected resembled faeces, being similar in size, shape, and moisture content. To avoid cross-contamination between sites the digesta was squeezed directly into individual sterile containers. Samples were then stored at −20 °C prior to molecular work.

DNA extraction and 16S rRNA gene library preparation

DNA extraction was performed using the QIAamp® Fast DNA stool mini kit from QIAGEN according to the manufacturer’s protocol (Qiagen Pty. limited Victoria, Australia). Duplicate samples from each GI region for both wombat species were processed. BNW samples were extracted at the University of Tasmania, and SHNW samples were extracted at the University of Adelaide. BNW DNA extracts were then shipped frozen to the University of Adelaide for library preparation. Barcoded V4 region 16S rRNA gene amplicons were generated using primers from Caporaso et al. (2011) (forward primer 515F: GTGCCAGCMGCCGCGGTAA and barcoded reverse primer

806R: GGACTACHVGGGTWTCTAAT). The PCR reactions were prepared in a pre-PCR laboratory in a 5% sodium hypochlorite-cleaned and UV irradiated hood. Single reactions (Marotz et al., 2019) of 2.5 µL X10 HiFi buffer, 0.1 µL Platinum™ Taq DNA Polymerase (ThermoFisher), 19.2 µL dH2O, 0.2 µL 100 mM dNTP mix, 0.5 µL each of 10 µM forward and reverse primer and 1 µL input DNA. The DNA was amplified using an initial denaturation at 94 °C for 3 min, followed by 35 cycles of denaturation at 94 °C for 45 s, annealing at 50 °C for 1 min, elongation at 68 °C for 90 s, with final adenylation for 10 min at 68 °C (Thompson et al., 2017). Gel electrophoresis was performed on PCR reactions on a 3.5% agarose gel to ensure samples contained amplicons of the desired length (∼390 bp). For each sample, 1 µL PCR amplified DNA was mixed into 199 µL Qubit® working solution (diluted Qubit® dsDNA HS Reagent 1:200 in Qubit® dsDNA HS Buffer) and quantified using a Qubit® 2.0 Fluorometer. Samples were then pooled equimolar and cleaned using AxyPrep™ (Axygen) following the manufacturer’s instructions. The final pool was quantified and quality checked using an Agilent TapeStation. DNA sequencing was performed on an Illumina MiSeq (v2, 2 × 150 bp) at SAHMRI (South Australian Health and Medical Research Institute).

Bioinformatic analyses

QIIME2 (Bolyen et al., 2019) and R v4.0.2 (R Core Team, 2020) were used to perform the bioinformatic analyses and create figures. Reproducible code is publicly available at the following GitHub repository (https://github.com/EisenRa/2021_Wombat_GI_tract_16S). Forward reads were denoised using deblur (Amir et al., 2017) in QIIME2. Representative sequences were assigned taxonomy using the QIIME2 feature-classifier plugin (Bokulich et al., 2018) on the pre-trained SILVA (Quast et al., 2013) 138 V4 region classifier. A phylogenetic tree was created using the SATé-enabled phylogenetic placement (SEPP) technique (Janssen et al., 2018) using the fragment-insertion QIIME2 plugin. Data was then imported into rStudio into a phyloseq (McMurdie & Holmes, 2013) object using qiime2R (https://github.com/jbisanz/qiime2R), and manipulated with dplyr (https://github.com/tidyverse/dplyr), tidyr (https://github.com/tidyverse/tidyr/), and the microbiome package (https://github.com/microbiome/microbiome/). For all sample type comparisons, the table was rarefied at a depth of 21,710, for colon-only comparisons the table was rarefied to 47,301. ASVs that were found in only 1 single replicate of a gut region were removed. These ASVs represented only 4.3% of identified ASVs and 0.16% of the mean relative abundance of the dataset. Alpha and beta diversity measures (including UniFrac (Lozupone & Knight, 2005)) were calculated in R using phyloseq, and figures were plotted using ggplot2 (Wickham, 2016) and ggVennDiagram (Gao, Yu & Cai, 2021). Compositional analysis of the community was done using ANCOM-II in R (ANCOM v2.1 (Kaul et al., 2017) in R v.4.0.3). Analysis was undertaken using an ANCOM - adjusted for covariates model which accounts for the covariation expected at the species/animal level. The analysis was performed at the rarefied genus level for the proximal colon and distal colon samples only. For the venn and euler diagrams, ASVs needed more than 2 reads assigned at a given site (between two replicates) to be considered detected, and were then plotted using ggVenDiagram (Gao, Yu & Cai, 2021).

Results

DNA sequencing of the 30 gastrointestinal digesta samples (Fig. 1) and 1 extraction blank control (EBC) resulted in 4,923,858 (mean of 158,834 ± 76,389) forward reads (R1), which were denoised using deblur into 2,524 amplicon sequence variants (ASVs).

Microbial biogeography through the digestive tract of two wombat species

To test if these different gut regions influenced the microbial communities, we analyzed the microbial composition, richness (the number of ASVs) , and diversity of duplicate luminal samples taken throughout the GI tract of each wombat species. We first investigated what types of microbes (the taxonomic composition) were present in the different GI samples for both wombat species. At the phylum level, stomach and small intestine samples tended to be dominated by Proteobacteria, Firmicutes, and Fusobacteriota (Fig. S1) We observed a large difference in phyla composition between the proximal (PSI) and distal small intestine (DSI) samples from the SHNW, with the PSI dominated by Proteobacteria, and DSI dominated by Fusobacteria (Fig. S1). The dominant families in the stomach and small intestine samples were Pasteurellaceae (59% in BNW-ST (ST = Stomach), 56% in SHNW-ST, 43% in SHNW-PSI), Peptostreptococcaceae (67% in BNW-SI), and Fusobacteriaceae (81% in SHNW-DSI) (Fig. 2). In contrast, the colon samples for both wombat species were dominated by the Bacteroidota, Firmicutes, and Spirochaetota phyla (Fig. S1). We also observed taxonomic differences between proximal (especially proximal colon 1&2) and distal colon samples for both the wombat species (Fig. 2). The first proximal colon samples were dominated by Prevotellaceae (∼40% relative abundance), and Fibrobacteraceae (∼10% for the BNW), whereas the latter proximal colon and distal colon samples contained higher proportions of Spirochaetaceae, Rikenellaceae, and WCHB1-41. A heatmap of all microbial families is available in Fig. S4.

Figure 2 Taxonomic composition of the different sample types at the family level.

Only the top 20 most abundant families are displayed for clarity (these families account for 81.17% of reads). The ‘Other’ bin (grey) contains the other families. Replicate samples were merged per sample site. (A) bare-nosed wombat. (B) southern hairy-nosed wombat. The widths of the bars are scaled to the length of the GI region.

Microbial richness tended to increase through the length of the digestive tract for both wombat species (Fig. 3). Stomach and small intestine samples had the lowest ASV richness (∼100–200), followed by the first two proximal colon samples (PC1/PC2: ∼400–500), with the latter proximal colon and distal colon samples exhibiting the highest ASV richness (∼500–800). Due to our limited sampling, we were unable to run linear models to confirm this pattern statistically, and future studies employing larger sample sizes should test the observed pattern.

Figure 3 Microbial diversity (ASV richness) of samples collected throughout the wombat digestive tract, ordered from start to end.

ST, stomach; SI, small intestine; PSI, proximal small intestine; DSI, distal small intestine; PC, proximal colon; DC, distal colon. Two technical replicates were collected and processed for each site (joined by lines). (A) Bare-nosed wombat. (B) Southern hairy-nosed wombat. Digestive tract drawings were adapted from Barboza & Hume (1992a).

Like richness, the types of microorganisms living in a site and how they are structured (composition) can also be influenced by the environmental factors of that site. To test this, we compared the microbial community structure of these distinct sites using both the abundance weighted and unweighted UniFrac distance metrics. As predicted, major differences were observed in microbial composition along the GI tract in both wombat individuals, with stomach, small intestine colon, and colon samples clustering across the axis of most variation (Axis 1: Figs. 4A/4B). Note that the colon samples are likely driving the placement of samples on the first two axes. Colon samples from the different wombats were separated across axis 2 for the unweighted UniFrac analysis (Fig. 4A), and separation between proximal and distal colon samples was observed for both unweighted and weighted UniFrac distances (Fig. 4). Interestingly, the microbial composition of the first proximal colon samples of different wombat species is more similar to each other than they are to the distal colon samples of the same species (Fig. 4B, Fig. S2B). Considering colon samples only, potentially species-specific differences in abundance-weighted microbial composition between distal colon samples were observed across axis 2 (Fig. S2B). Note however, that we only have 1 sample per species, and can therefore not account for intra-species diversity. These results indicate that microbial diversity and composition vary throughout the GI tract of two hindgut fermenting species, and that the start of the proximal colon—the putative primary site of fermentation in these wombat species—is distinct from the distal colon.

Figure 4 Microbial community structure among sites along the wombat GI tract.

(A) PCoA of unweighted UniFrac distances and (B) PCoA of weighted UniFrac distances. Samples are coloured by sample type, and shaped by host species. Numbers indicate order in which samples occur in the digestive tract. Lines connect samples from the same sample type.

Microbial differences between proximal and distal colon sites

Because the proximal colon is the putative primary site of plant fermentation in wombats, we next focused on further characterising microbial community compositional differences between the proximal and distal colons for both species. As there were many ASVs that were classified to the same taxa we collapsed the ASV table to the genus-level, and ran ANCOM-II to identify genera that were differentially abundant between different regions of the colon (i.e., PC1, PC2, DC, etc.). Of the 253 genera classified in the dataset, ANCOM-II identified 60 that were significantly differentially abundant throughout the colon when the W-statistic threshold for rejecting the null was at 70% (Fig. 5) (Table S1). Taxa that had significantly higher abundance in PC1/2 vs. DC include: Prevotella (W = 245), Prevotellaceae_UCG-001 (W = 241), Bacteroides (W = 239), and Bacteroidales_RF16_group (W = 231). Taxa with a higher relative abundance in DC vs. PC1/2 include: Bacteroidales_BS11_gut_group (W = 250), Bacteroidales_UCG-001 (W = 243), WCHB1-41 (W = 242), and Izemoplasmatales (W = 226).

Figure 5 Heat map of the microbial genera that were found to be differentially abundant between proximal and distal colon sites.

The abundance of genera that were found to be differentially abundant (0.7 threshold) in the ANCOM-II analysis are displayed. Black indicates 0 assigned reads. Non-genus assignments are prefixed with the lowest level of taxonomy that could be assigned (e.g., f– = family). Taxonomy string are prefixed with ‘Phylum –’.

To test how representative the microbial community of faeces are to the first proximal colon site, we measured how many ASVs were shared between PC1 and the last DC sample for each wombat species. Surprisingly, only 98 (10.6%) and 206 (18.8%) ASVs were shared between PC1 and DC samples for the BNW and SHNW, respectively (Fig.6). See Fig. S3 for a cross-species comparison. The ASVs that were unique to the BNW PC1 and DC sites accounted for 25% and 64% of the relative abundance in those sites, respectively. Likewise, the ASVs that were unique to the SHNW PC1 and DC sites accounted for 32% and 67%, respectively. These differences remained after collapsing ASVs to the genus-level (Fig. S5), albeit with a smaller magnitude (relative abundance of site-specific genera: BNW DC = 5% BNW PC = 20% SHNW DC = 3% SHNW PC = 12%). These results indicate that the first proximal colon and distal colon sites harbour distinct communities of microbes.

Figure 6 Euler diagram of amplicon sequence variants (ASVs) shared between proximal and distal colon sites for both wombat species.

Euler diagram of amplicon sequence variants (ASVs) shared between proximal colon 1 (PC1) and last distal colon site (DC) for (A) bare-nosed wombat (BNW) and (B) southern hairy-nosed wombat (SHNW). Percentages represent the proportion of ASVs specific to a given area.

Discussion

Mammalian gut microbial communities play vital roles in the harvesting of energy and nutrients (Rowland et al., 2018). Due to ease of sampling, most previous mammalian gut microbiota research has focused on faecal samples as proxies for the gut microbiota. However, the mammalian GI tract contains a series of distinct microbial growth conditions that can structure the diversity and composition of resident microbial communities (Donaldson, Lee & Mazmanian, 2015). Outside of model or domesticated placental mammal species there has been little work in characterising how GI microbial communities are structured along GI tracts, and how representative faeces are of other GI regions. Our work here fills this gap for two species of free-living, large, hindgut-fermenting marsupials—the bare-nosed wombat (BNW) and the southern hairy-nosed wombat (SHNW). Although we had data for only two individual wombats, our results showed that microbial community composition, structure and diversity varied along the wombat GI tract , and that the first proximal colon region—the putative primary site of fermentation—was highly dissimilar to the distal colon. Our findings also indicate that faecal samples collected for wombats may not be representative of the primary site of fermentation. Future studies seeking to understand the roles that GI microbial communities play in host energy acquisition and health should consider the microbial biogeography of the mammalian GI tract.

A major finding was that the microbial communities present in the first proximal colon region (PC1) were more similar between both individuals, than to subsequent colon region samples from the same individual. This is particularly interesting given that both individuals are from different species—though still preliminary given the small sample size in our study (n = 1 for both species). PC1 in both wombat individuals were dominated by the family Prevotellaceae (∼40% relative abundance), a family that contains members with extensive carbohydrate/fiber fermentation capabilities (Filippo et al., 2010; Kovatcheva-Datchary et al., 2015). All other GI sites contained lower levels of Prevotellaceae (<5% relative abundance). In addition, PC1 for both wombat individuals contained significantly higher levels of bacteria classified to the genus Bacteroides, which are known to possess a diverse suite of carbohydrate-active enzymes (Kaoutari et al., 2013). PC1 of the BNW had relatively high levels (∼10% relative abundance) of the genus Fibrobacter (∼0.3% in the SHNW PC1), which have known fiber-degrading capacity and have been found in various placental foregut- and hindgut-fermenting herbivores (Neumann, McCormick & Suen, 2017). These distinct microbial compositions in PC1 of both wombat individuals compared to subsequent colon regions warrant further investigation with shotgun metagenomics to verify whether these taxonomic differences represent differences in the capacity to ferment fibre. This would further test Barboza and Hume’s finding that PC1 represents the primary site of microbial fermentation in the wombat GI tract, and offer us a greater understanding of microbial fermentation in wombats. Finally, such a study could also verify whether the high proportion of Firmicutes found in colon samples are involved fibre degradation as expected (Biddle et al., 2013).

Given the major microbial compositional differences observed between proximal and distal colon samples, we sought to determine how many proximal colon-associated microbes would be detected by sampling faeces, the most commonly used sample type in mammalian gut microbiota research. Surprisingly, only a very small proportion of PC1 ASVs could be detected in distal colon samples (10.6% and 18.8% in the BNW and SHNW, respectively). This suggests that seeking to understand microbial functions at the putative primary site of fermentation through faecal samples may not be feasible. This is not to say that faecal samples are not useful, as previous gut microbiome research on captive (Shiffman et al., 2017) and wild wombats (Eisenhofer, Helgen & Taggart, 2021; Weiss et al., 2021) have yielded insights into wombat digestion and ecology. Eisenhofer et al. used faecal samples from both captive and wild SHNWs to show that captivity has a large influence on the faecal microbiota of SHNWs. They were also able to identify population-specific microbial signatures and found a correlation between habitat type (degraded vs. intact) and faecal microbiota composition in SHNWs (Eisenhofer, Helgen & Taggart, 2021). (Shiffman et al. (2017) used shotgun metagenomic sequencing on faecal samples collected from a captive SHNW, and found numerous microbial genes involved in plant degradation and urea recycling. Interestingly, they made note that they could not detect bacteria from the phylum Fibrobacterota, which are commonly found in herbivorous mammals (Neumann, McCormick & Suen, 2017). We identified Fibrobacterota in both wombat species, with a higher relative abundance in the BNW, particularly in the proximal colon. Overall, our results suggest that faeces may not be a representative sample type for the putative primary site of microbial fermentation in wombats. We propose that future research seeking to understand the roles that GI microbes play in wombat (or other mammal) digestion consider this, and aim to study other regions of the GI tract where possible.

Our study is not without limitations. The opportunistic nature of sample collection from fresh necropsied animals limited this study to only one animal for each wombat species. This hampered our ability to compare microbial differences between wombat species, as the differences observed may be related to individual-level variation—making us unable to run statistical tests on GI tract alpha or beta diversity. However, the consistency observed between the biological replicates and in overall trends for both species of wombat support our findings relating to PC1/distal microbial differences and the level of representability of the faeces to PC1—despite the ecological differences between wombat species and the estimated ∼8 million years since they last shared a common ancestor (Mitchell et al., 2014). Another limitation is that we could not distinguish DNA from living or dead cells, and it is possible that the differences between GI sites is greater than measured in our study due to dead cells/relic DNA moving from site to site and inflating similarity (Lennon et al., 2018). Future studies with larger sample sizes will enable more detailed microbial comparisons between wombat species, and improve estimates of within-animal microbial differences along the GI tract.

The findings of our study have spawned several promising avenues of future research. Using shotgun metagenomics to compare and contrast the microbial functions present between proximal and distal colons could allow for a greater understanding of the roles GI microbes play in wombat digestion and health. The strain-level information obtained from shotgun metagenomics would also allow for greater tracking of microbes along the GI tract, and help disentangle issues relating to the over splitting of microbes due to intragenomic heterogeneity in 16S rRNA genes (Sun et al., 2013). Future, larger sample size comparisons between wombat species could also seek to identify microbes that are shared between wombat species, and consistently found within species. Any such ‘core’ microbes that are still shared between wombat species despite ∼8 million years of separation could indicate that they are important for host health. In our study the distal colon exhibited the greatest microbial differentiation between wombat species. Such interspecies microbial differences in the latter parts of the colon could be due to GI morphological/physiological differences between wombat species (Barboza & Hume, 1992a) and relate to differences in the ecology of the two wombat species. Further larger-scale comparisons between BNW (mesic living) and SHNWs (arid/semi-arid living), and other phylogenetically distinct mammals could offer insights into the potential roles that GI microbes play in the arid and temperate adaptability of mammals in Australia.

Conclusions

Our study suggests that microbial communities in the first proximal colon region—the putative site of primary plant fermentation in wombats—are distinct from the distal colon, and that faecal samples may have limitations in capturing the diversity of these communities. While faeces are still a valuable and effective means of characterising the distal colon microbiota, future work seeking to better understand how GI microbes impact the energy economy of wombats (and potentially other hindgut-fermenting mammals) may need to take gut biogeography into account.

Supplemental Information

Figure S1 Taxonomic composition of the different sample types at the phylum level

Taxonomic composition of the different sample types at the phylum level. Only the top 10 most abundant phyla are displayed for clarity (these phyla account for >99% of reads). Replicate samples were merged per sample site. (A) bare-nosed wombat. (B) southern hairy-nosed wombat. The widths of the bars are scaled to the length of the GI region.

Click here for additional data file.

Figure S2 PCoA of colon samples only

Ordination of colon samples only. (A) PCoA of unweighted UniFrac distances and (B) PCoA of weighted UniFrac distances. Samples are coloured by sample type, and shaped by host species.

Click here for additional data file.

Figure S3 Venn diagram of ASVs shared between PC1 and DC for both species of wombat

Click here for additional data file.

Figure S4 Heat map of the microbial families at different regions of the wombat gastrointestinal tract

Black indicates 0 assigned reads. Non-family assignments are prefixed with the lowest level of taxonomy that could be assigned (e.g. o– = family). Taxonomy string are prefixed with ‘Phylum –’.

Click here for additional data file.

Figure S5 Euler diagram of genera shared between proximal and distal colon sites for both wombat species

Euler diagram of genera shared between proximal colon 1 (PC1) and last distal colon site (DC) for (A) bare-nosed wombat (BNW) and (B) southern hairy-nosed wombat (SHNW). Percentages represent the proportion of genera specific to a given area.

Click here for additional data file.

Table S1 ANCOM results at the genus level

ANCOM results for the rarefied genera found significantly differentially abundant between the hind-gut sites at a W-statistic threshold at 0.7.

Click here for additional data file.

The authors would like to thank Alejandro Correa Lozano for his help collecting and analysing the BNW samples.

Abbreviations

ANCOM Analysis of Composition of Microbiomes

ASV Amplicon Sequence Variant

BNW Bare-Nosed Wombat

DC Distal Colon

DSI Distal Small Intestine

GI Gastrointestinal

PC Proximal Colon

PCoA Principal Coordinates Analysis

PCR Polymerase Chain Reaction

PSI Proximal Small Intestine

SI Small Intestine

SHNW Southern Hairy-Nosed Wombat

ST Stomach

Additional Information and Declarations

Competing Interests

Author Contributions

Animal Ethics

DNA Deposition

Data Availability

The authors declare there are no competing interests.

Raphael Eisenhofer and Erin D’Agnese conceived and designed the experiments, performed the experiments, analyzed the data, prepared figures and/or tables, authored or reviewed drafts of the paper, and approved the final draft.

David Taggart and Scott Carver conceived and designed the experiments, authored or reviewed drafts of the paper, and approved the final draft.

Beth Penrose conceived and designed the experiments, prepared figures and/or tables, authored or reviewed drafts of the paper, and approved the final draft.

The following information was supplied relating to ethical approvals (i.e., approving body and any reference numbers):

NA, samples were opportunistically scavenged.

The following information was supplied regarding the deposition of DNA sequences:

The demulitplexed DNA sequencing data are available at NCBI SRA: BioProject ID PRJNA727080.

The following information was supplied regarding data availability:

All QIIME2, code, analysis files, and R code used for analyses and to plot figures are available at GitHub: https://github.com/EisenRa/2021_Wombat_GI_tract_16S.

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
