# Peer review of "Microbial biogeography of the wombat gastrointestinal tract"

_PeerJ, doi:10.7717/peerj.12982_

## Round 0.1 · original submission · Major Revisions

Dear Dr. Eisenhofer and colleagues:

Thanks for submitting your manuscript to PeerJ. I have now received three independent reviews of your work, and as you will see, one reviewer recommended rejection, while the others suggested minor revisions (with many suggested changes). I am affording you the option of revising your manuscript according to all three reviews.

Please note that reviewer 2 has included a marked-up version of your manuscript.

You provide lots of data on two individuals but this prevents robustly testing the hypothesis that species differ. The finding that microbes differ along the digestive tract is not surprising (microbiomes tend to be very plastic across individuals let alone populations) and needs qualification from measures of microbial substrates and microbial environment.

There are many other problems pointed out by the reviewers, and you will need to address all of these and expect a thorough review of your revised manuscript by these same reviewers.

Therefore, I am recommending that you revise your manuscript, accordingly, taking into account all of the issues raised by the reviewers.

I look forward to seeing your revision, and thanks again for submitting your work to PeerJ.

Good luck with your revision,

-joe

·

Basic reporting

The manuscript by Eisenhofer et al. seeks to compare the microbiota of different gut regions in two species of wombat. The paper is generally well written, and the findings clearly articulated.

However, I could not see reference to where the raw data for this study has been deposited in the manuscript itself. Please add this to the paper before publication.

Minor comments:

L64-72: There has been a study that looked at the different GI sites of the koala (Barker, Christopher J., et al. "Investigation of the koala (Phascolarctos cinereus) hindgut microbiome via 16S pyrosequencing." Veterinary microbiology 167.3-4 (2013): 554-564.). Given that the koala and wombat and phylogenetically closely related species, it seems remise not to outline this study here and compare its findings to those of this study in the discussion.

L197: If I am reading Figure S1 right (I do not have the caption for this figure) then it seems that only the distal colon samples really separate based on species on axis 2 of B. This seems to be indicated in the next sentence, however, I would suggest making that clear here.

L206-210: Do you mean figure S2 here?

Figure 1B: DC1 and DC3 are labelled but there is no DC2. Is this a typo or is there another reason for this?

Figure 3: I would like to see a larger number of families represented in this plot as for the DC in the SHNW a large proportion of the microbiome is assigned to “other” making it difficult to see the detail of this community.

Experimental design

The main weakness of the paper is that only a single animal of each species was studied making it difficult to know if the between species differences found were true species differences or inter individual variation. However, this weakness is acknowledged by the authors in the discussion and given the difficulty in obtaining such samples the results are still worth reporting.

There are also a few instances where further information on the methods would be valuable as outlined in my specific comments.

Specific comments:

L163: Please clarify, so the ASV level analysis was or was not performed on the rarefied dataset?

Figure 1: Is it possible to indicate exactly where in the GI tracts the samples were taken from? For instance, the small intestine is quite long in both species. Were the samples collected from the middle of this section or towards one end? This may be important for reproducibility of the findings.

Validity of the findings

My main concern is related to the preliminary data manipulation prior to the analyses. Specifically, the exclusion of ASVs found in a single sample. While these ASVs will not give you any information about sharing among samples they may provide important information on the individuality of samples and on alpha diversity. The removal of singltons (ASVs for which only a single read is returned) is common and reasonable as they often represent sequencing error but the removal of a sequence because it is only seen in a single sample could be misleading. If my interpretation of how the analyses were performed is correct then this will artificially reduce the variation between samples, making them appear more homogeneous. Reducing variation in this way could make differences between the GI sites appear more convincing. Therefore, I think it is important for all ASVs to be included in the analyses or at the very least for the authors to show (in an online supplement) that it does not change the findings.

Minor comments:

L187-200: When I look at figure S2 it seems that the small intestine samples are actually quite distinct between the two species and also that the two SI sites in the SHNW are quite different from one another, which does not come across in the PCoA. Perhaps the PCoA is mostly reflecting the differences in richness between the different regions? I think it is important to identify this in the text.

L258-274: The distal colon had a high proportion of Firmicutes and this phylum has many fibrolytic members. Can you comment on how the overall capacity for fermentation is likely to differ between the GI sites based on the microbial composition? While no doubt the PC is a site of significant microbial fermentation I am not convinced that the distal colon is not based on the discussion of the results presented here.

Figure 1: Given there are only two or three samples per gut region, boxplots that show the different quartiles are not an appropriate way to represent the data. Instead, please just plot the raw data points without the boxplots.

Additional comments

Minor comments:

L232-240: I would be really interested to see what the overlap is between the sites if you repeat this analysis at the genera or family level. Given the known issues with intragenomic heterogeneity in 16S genes and also in assigning ASVs to lower taxonomic levels the differences seen in composition may not be biologically meaningful.

·

Basic reporting

The manuscript is well-written and communicates a clear story based on the study data. I had some comments where the authors did not include important relevant details, and made suggestions for improvement (see attached file). I also suggested that the authors change the order of findings in the results section so that is easier to follow.

Experimental design

I made minor suggestions for communicating study aims and predictions more clearly. The authors do need to be more transparent about their sampling and study limitations throughout their manuscript. They do not need to dwell on this, they just need to remind readers that their data comes from only two individuals, and thus, their inference (and statistical power) is limited. Again, see my comments on the attached file.

Validity of the findings

The methods section was detailed and the research could be reproduced by others. The sequence data and code are accessible. The conclusions are tied to the study objectives.

Additional comments

Please see the attached PDF file for my detailed feedback and suggestions to strengthen the manuscript.

Reviewer 3 ·

Basic reporting

The authors present new observations of microbial diversity using current methods with molecular markers. The authors report that microbial diversity differs between the regions of the digestive tract. Most importantly assessments of digesta in the terminal part of the tract may be distinct from those taxa in the primary fermentative region in the proximal colon.

Experimental design

The interpretation of the data is limited by the lack of observations on the substrates available to microbes (e.g, plan fiber fractions, plant taxa) and the environment in the tract (e.g., pH, osmolality, temperature). The flow of substrates to the microbes and the turnover of the medium in any section could be inferred from the state of the host (e.g., season, sex, reproductive state, age, body mass and body condition).

Validity of the findings

The study is best described as a preliminary observation because only one individual of each species was sampled. This experimental design confounds individual with species, that is, the design is not sufficient to compare species without providing more observations of individuals within each species. It is likely more samples can be obtained from wild specimens that are killed on the roads for SHNW and BNW.

Additional comments

Line 178, "is thought to occur" seems overly tentative since fermentation rates and the digestion of plant fiber was reported for both species.
Line 193. species and indvidual are confounded because there is only one individual per species.
line 205 "At the phylum level, these samples" Do you mean stomach and small intestine samples?
Line 243. How does this work help us to understand the relationship between miocrobe and host? What is the big picture here? See recent papers
R. Kartzinel, J. C. Hsing, P. M. Musili, B. R. P. Brown, R. M. Pringle, Covariation of diet and gut microbiome in African megafauna. Proceedings National Academy of Sciences USA 116, 23588-23593 (2019).
C. A. Rojas, S. Ramirez-Barahona, K. E. Holekamp, K. R. Theis, Host phylogeny and host ecology structure the mammalian gut microbiota at different taxonomic scales. Anim Microbiome 3, 33 (2021).

Line 282. Both animals died after an illness. Intake of food and water was likely interrupted. Diet is also different. There is not enough replication to compare species just the samples within indviduals.
These data are not sufficient to test the hypothesis that fecal and colonic microbes differ. It is possible but you need more indviduals with covariates of diet composition.
Line 301. "However, …" is a long compliicated sentence. What are biological replicates? Are these observations of multiple samples from the same region? Consistency in PC scores is a circular argument because the scores are the outcome of partitioning variation in the data set with the statistical analysis.
Line 307. Any literature to support this idea about ned and live cells?

---

## Round 0.2 · Minor Revisions

Dear Dr. Eisenhofer and colleagues:

Thanks for revising your manuscript. The two reviewers are very satisfied with your revision (as am I). Great! However, there are some concerns raised by both reviewers. Please address these and submit a revision ASAP.

Best,

Good luck with your revision,

-joe

·

Basic reporting

No comment

Experimental design

No comment

Validity of the findings

No comment

Additional comments

I have reviewed a previous version of this manuscript and in general the authors have adequately addressed my comments and concerns. There are, however, a few points where I feel further improvements could be made as detailed below.

I appreciate the authors taking the time to perform the additional analyses I requested. I feel that there would be benefit to having analysis 2 and 3 included as part of the supplementary documents and referenced in the main text. However, if the authors make the additions I suggest on L170 of the revised manuscript then analysis 1 is probably not necessary to present.

L170: I suggest adding the information presented in the response to my comment here to reassure other readers that the removed ASVs were only a small portion of the dataset. i.e “These ASVs represented ~4.3% of identified variants and 0.16% of the average relative abundance of the dataset.” I also suggest clarifying that ‘sample’ here refers to a single replicate of a gut region.

L214: I suggest adding a sentence after the note here (or at another relevant point in the m.s.) pointing out the differences between SI regions in the SHNW. e.g. “On further examination of the relative abundance of the different Phyla across gut regions (supplementary figure 2) a large difference between the proximal and distal small intestine samples from the SHNW was observed, with PSI dominated by Proteobacteria and DSI dominated by Fusobacteria.”

L280-283: I would like to see a sentence here that acknowledges the high proportion of Firmicutes in the DC and their potential involvement in fibre degradation. https://www.mdpi.com/1424-2818/5/3/627 is a good reference for this.

Figure 5 and New supplementary heatmap: I think the addition of the supplementary heatmap showing all the families would be a good addition to the current set of supplementary figures. I would, however, like to see the higher-level taxonomy added to both heatmaps, particularly as some ASVs are undefined at the family level or have uninformative names such as “uncultured”.

·

Basic reporting

Manuscript is detailed and well-written, and contains sufficient background and interpretations. See minor comments in the “General comments” section.

Experimental design

Methods are reproducible and raw data/code are publicly available.

Validity of the findings

Minor comments to improve clarity, and comments to authors to be cautious when mentioning “species differences” of any kind as they do not have the statistical power to do so.

Additional comments

Introduction
L34- replace “For both the bare-nosed wombat and the southern hairy-nosed wombat” with “For both wombat individuals”

L99-101: rephrase to “We predicted that the proximal colon, the primary site for plant fermentation, would harbor a distinct microbial community […]”

Results
L193-unclear what the acronyms stand for
L192- in supplementary material, this figure is labeled as SFig2 instead of SFig1

L200- would add that for both wombat individuals, compositional differences are more apparent between PC1&2 and the latter proximal colon and distal colon.

L207- start new paragraph with the sentence “Like richness”, to show authors are transitioning to beta diversity analysis

L209-210 - rephrase “we calculated the microbial compositions of these distinct sites” to “we compared the microbial community structure of these distinct sites”

L211-212- rephrase “Using both methods, the major differences observed in microbial composition were associated with GI tract site not wombat species, with stomach” to “As predicted, major differences were observed in microbial composition along the GI tract in both wombat individuals, with stomach…” I do not believe it is appropriate to mention host species analyses in this sentence.

L214-217- please clarify the sentence. As it currently reads, it is saying the same thing twice

L217-222: move these two sentences after the sentence “Interestingly, the microbial composition of the first proximal colon.”

L244- SI figure 3 to SI Figure 3

L269-285: authors can talk about the “species” differences without having to mention the word “species.” The authors do not have the statistical power to discuss "species" differences because they have an N=1 for each species. Thus, authors can instead say: “A major finding was that the microbial communities present in the first proximal colon region (PC1) were more similar between wombat individuals than to subsequent colon region samples from the same individuals”. Authors can still mention why the inter-individual differences were observed (the two wombat individuals represent distinct species that differ in their gut morphologies and habitats, etc).

L342- authors can add that future work can be comparative and include more host species that are closely and distantly related to the wombat to look at the potential role of host phylogeny as well.

Supplemental
I could not find the legends for the supplemental figures again
Include a legend in the spreadsheet for Table S1

Figures
Fig4- reorder color key so that stomach appears first, then small intestine, proximal colon, etc
Fig6- use non-acronyms whenever possible, I suggest authors spell out full species name (Bare-nosed wombat) at the top of the diagrams, and then for example shorten BNW.PC1 to just PC1

---

## Round 0.3 · accepted · Accept

Dear Dr. Eisenhofer and colleagues:

Thanks for revising your manuscript based on the concerns raised by the reviewers. I now believe that your manuscript is suitable for publication. Congratulations! I look forward to seeing this work in print, and I anticipate it being an important resource for groups studying wombat biology and evolution. Thanks again for choosing PeerJ to publish such important work.

Best,

-joe